# Reliability analysis of offshore wind turbine foundations under lateral cyclic loading

Gianluca Zorzi[1], Amol Mankar[2], Joey Velarde[3], John D. Sørensen[2], Patrick Arnold[1], Fabian Kirsch[1]

[1] GuD Geotechnik und Dynamik Consult GmbH, Berlin, 10589, Germany

[2] Aalborg University, Aalborg, 9100, Denmark

[3] COWI A/S, Aarhus, 8000, Denmark

*Correspondence to*: Gianluca Zorzi (zorzi@gudconsult.de)

**Abstract.** The design of foundations for offshore wind turbines (OWT) requires the assessment of long-term performance of the soil–structure-interaction (SSI), which is subjected to many cyclic loadings. In terms of serviceability limit state (SLS), it

has to be ensured that the load on the foundation does not exceed the operational tolerance prescribed by the wind turbine manufacturer throughout its lifetime. This work aims at developing a probabilistic approach along with a reliability framework with emphasis on verifying the SLS criterion in terms of maximum allowable rotation during an extreme cyclic loading event. This reliability framework allows the quantification of uncertainties in soil properties and the constitutive soil model for cyclic loadings and extreme environmental conditions and verifies that the foundation design meets a specific target reliability level.

A 3D finite element (FE) model is used to predict the long-term response of the SSI, accounting for the accumulation of permanent cyclic strain experienced by the soil. The proposed framework was employed for the design of a large diameter monopile supporting a 10 MW offshore wind turbine.

## 1 Introduction

Offshore wind turbines are slender and flexible structures which have to withstand diverse sources of irregular cyclic loads

(e.g., winds, waves, and typhoons). The foundation, which has the function to transfer the external loads to the soil must resist this repeated structural movement by minimising the deformations.

The geotechnical design of a monopile foundation for an OWT has to follow two main design steps named static load design (or pre-design) and cyclic load design. A design step is mainly governed by limit states: i.e. the ultimate limit state (ULS), the serviceability limit state (SLS) and the fatigue limit state (FLS). The design of an offshore structure mostly starts with the

static load design step in which a loop between the geotechnical and structural engineers is required to converge to a set of optimal design dimensions (pile dimeter, pile length and can thickness). This phase is governed by the ULS in which it must be ensured that the soil's bearing capacity withstands the lateral loading of the pile within the allowable deformations (i.e. pile deflection/pile rotation at the mudline).

Subsequently, the pre-design is checked for the cyclic load scenario (cyclic load design). In this step, the verification of the pre-design regards three limit states: ULS, SLS and FLS. The cyclic stresses transferred to the soil can reduce the lateral resistance by means of liquefaction (ULS), change of the soil stiffness which can cause resonance problems (FLS) and progressively accumulate deformation into the soil leading to an inclination of the structure (SLS). If one of these limit states is not fulfilled, cyclic loads are driving the design and the foundation dimensions should be updated.

Performing the checks for the cyclic load design step is very challenging due to the following: (i) a high number of cycles is usually involved; (ii) soil subjected to cyclic stresses may develop at the same time non linearity of the soil response, pore water pressure, changing in stiffness and damping and accumulation of soil deformation (Pisano, 2019).; (iii) the load characteristic such as frequency, amplitude and orientation are continually varying during the lifetime; (iv) characteristic of the soil such as type of material, porosity, drainage condition can lead to different soil response; (iv) the relevant codes (BSH,

2015; DNV-GL, 2017) do not recommend specific cyclic load methods for predicting the cyclic load behaviour of structures which lead to the development of various empirical formulations (Cuéllar, et al., 2012, Hettler, 1981, LeBlanc, et al., 2009) or numerical based models (Zorzi et al 2018, Niemunis, et al., 2005, Jostad, et al., 2014, Achmus et al, 2007). Despite the different techniques used in these models, they all predict the soil behaviour "explicitly", based on the number of cycles instead of a time domain analysis (Wichtmann, 2016). Dealing with a high number of cycles, time domain analysis is not convenient due

to the accumulation of numerical errors (Niemunis, et al., 2005).

In common practice due to the non-trivial task faced by the engineers, simplifications and hence introduction of uncertainties and model errors, are often seen. The application of probabilistic-based methods for designing offshore foundations is not a new topic (Velarde et al., 2019a, Velarde et al., 2019b, Carswell et al.,2014); and it is mainly related to the static design stage. Very limited research has been developed regarding the probabilistic design related to the cyclic load design.


This current work focuses on the cyclic loading design stage and the verification of the serviceability limit state (SLS). During the design phase, the wind turbine manufacturers provide a tilting restriction for operational reasons. The recommended practice DNV-GL-RP-C212 (DNV-GL, 2017) provides the order of magnitude for the maximum allowed tilting of 0.25° throughout the planned lifetime. This strict verticality requirement may have originated from different design criteria, which

however, are mainly rooted within the onshore wind turbine sector and are given below (extracted from Bhattacharya, 2019):

•     Blade–tower collision: owing to an initial deflection of the blades, a possible tilting of the tower may reduce the blade–tower clearances.

•     Reduced energy production: change in the attack angle (wind-blades) may reduce the total energy production.

•     Yaw motors and yaw breaks: reducing motor capacity for yawing into the wind.

•     Nacelle bearing: a tilted nacelle may experience different loadings in the bearing, causing a reduction of their fatigue life or restrict their movements.

•     Variation in fluid levels and cooling fluid movement.

- P-δ effect: the mass of the rotor-nacelle-assembly is not aligned with the vertical axis and this creates an additional overturning moment in the tower, foundation, grouted connection, and the soil surrounding the foundation.

• Aesthetic reasons.

In SLS designs, extreme as well as relevant accidental loads, such as typhoons and earthquakes, should be accounted for as they can be design-driving loads. A very strict tilting requirement, i.e., 0.25°, in conjunction with these accidental conditions can increase the foundation dimensions and significantly raise the cost of the foundation.

In order to account for the behaviour of the soil-structure interaction (SSI) when subjected to high cyclic loads, an advanced
numerical method called soil cluster degradation (SCD) method was developed (Zorzi et al., 2018). This method explicitly predicts the cyclic response of the SSI in terms of the foundation rotation. The main objective of this study is to use the SCD method within a probabilistic approach. The probabilistic approach along with the reliability framework was used to quantify the main uncertainties involved in the SCD method (aleatoric and epistemic), explore which uncertainty the response is most sensitive to, and design the long-term behaviour of the foundation for a specific target reliability level. In this paper first the
developed reliability-based design (RBD) framework is outlined in detail. Finally, an application of the proposed RBD framework is presented for a large diameter monopile supporting a 10 MW offshore wind turbine.

## 2 Development of the RBD framework

### 2.1 Limit state function for SLS

The rotation experienced by the foundation structure subjected to cyclic loading is considered partially irreversible (irreversible
serviceability limit states) because the soil develops an accumulation of irreversible deformation due to the cyclic loading action. For this reason, it is noted that the accidental and environmental load cases for the SLS design are the extreme loads that give the highest rotation. As for a deterministic analysis, the first step in the reliability-based analysis is to define the structural failure condition(s). The term failure signifies the infringement of the serviceability limit state criterion, which is here set to a tilting of more than 0.25°. The limit state function $g(\boldsymbol{X})$ can then be written as

$$g(\boldsymbol{X}) = \theta_{max} - \theta_{calc}(\boldsymbol{X}) \,, \tag{1}$$

where $\theta_{max} = 0.25°$ is the maximum allowed rotation and $\theta_{calc}(\boldsymbol{X})$ is the predicted rotation (i.e. the model response) based on a set of input stochastic variables $\boldsymbol{X}$.

### 2.2 Estimation of the probability of failure

The design has to be evaluated in terms of the probability of failure. The probability of failure is defined as the probability of
the calculated value of rotation $\theta_{calc}(\boldsymbol{X})$ exceeding the maximum allowed rotation $\theta_{max}$ as it does when the limit state function $g(\boldsymbol{X})$ becomes negative, i.e.:

$$P_f = \mathrm{P}[g(\boldsymbol{X}) \leq 0] = P[\theta_{max} \leq \theta_{calc}(\boldsymbol{X})] \tag{2}$$

Once the probability of failure is calculated, the reliability index β is estimated by taking the negative inverse standard normal distribution of the probability of failure:

$\beta = \Phi^{-1}(P_f)$                            (3)

where $\Phi(\ )$ is the standard normal distribution function. The probability of failure in this work is estimated using the Monte-Carlo (MC) simulation. For each realisation, the MC simulation randomly picks a sequence of random input variables, calculates the model response $\theta_{calc}(X)$, and checks if $g(X)$ is negative (Fenton and Griffiths, 2008). Thus, for a total of $n$ realisations the probability of failure can be computed as:

$P_f = \frac{n_f}{n}$ ,                                (4)

with $n_f$ being the number of realisations for which the limit state function is negative (rotation higher than 0.25°).

IEC 61400-1 (IEC, 2019) sets as a requirement with regard to the safety of wind turbine structures, an annual probability of failure equal to $5 \times 10^{-4}$ (ULS target reliability level). This reliability level is lower than the reliability level indicated in the Eurocodes EN1990 for building structures where an annual reliability index equal to 4.7 is recommended. Usually, in the
Eurocodes, for the geotechnical failure mode considered in this paper the irreversible SLS is used. In EN1990 Annex B, an annual target reliability index for irreversible SLS equal to 2.9 is indicated, corresponding to an annual probability of failure of $2 \times 10^{-3}$.

IEC61400-1 does not specify the target reliability levels for the SLS condition. Therefore, it can be argued that the target for SLS in this paper should be in the range of $5 \times 10^{-4} - 2 \times 10^{-3}$. In this work, the same reliability target for ULS of $5 \times 10^{-4}$
is also considered for the irreversible SLS as a conservative choice.

## 2.3 Derivation of the model response $\theta_{calc}$

The calculation of the model response $\theta_{calc}$ is based on the soil cluster degradation (SCD) model. The SCD method explicitly predicts the long-term response of an offshore foundation accounting for the cyclic accumulation of permanent strain in the soil. The SCD model is based on 3D finite element (FE) simulations, in which the effect of the cyclic accumulation of
permanent strain in the soil is considered through the modification of a fictional elastic shear modulus in a cluster-wise division of the soil domain. A similar approach of reducing the stiffness in order to predict the soil deformation can be found in Achmus et al. (Achmus et al., 2007). The degradation of the fictional stiffness is implemented using a linear-elastic Mohr Coulomb model. Reduction of the soil modulus is based on the cyclic contour diagram framework (Andersen, 2015). The cyclic contour diagrams are derived from a laboratory campaign using cyclic test equipment. The tests are performed with different
combinations of cyclic amplitude and average load for N number of cycles. These diagrams provide a 3D relation between the stress level and number of cycles for an investigated variable: accumulation of strain, pore pressure, soil stiffness or damping. The cyclic contour diagrams have been applied successfully for many years for the design of several offshore foundations (Jostad, et al., 2014, Andersen, 2015), however careful engineering judgment is required for the construction and interpretation.

The loading input for the model must be a design storm event simplified in a series of regular parcels. This loading assumption is also recommended by DNV-GL-RP-C212 (DNV-GL, 2017) and the BSH standard (BSH, 2015). The method is implemented in the commercial code PLAXIS 3D (PLAXIS, 2017).

Three stochastic input variables ($X = [X_1, X_2, X_3]$) are necessary for the SCD model:

- $X_1$ = soil stiffness that is derived from the Cone Penetration Test (CPT);

- $X_2$ = cyclic contour diagram that is derived from the cyclic laboratory tests;

- $X_3$ = extreme environmental loads that are derived from metocean data and a fully coupled aero-hydro-servo-elastic model.

These inputs have to be quantified in terms of their point statistics (e.g., the mean, standard deviation, and probability distribution type) representing the uncertainties. When using the MC simulation, $100/p_f$ realisations are needed to estimate an accurate probability of failure, which makes it challenging to apply it in combination with the FE simulations. Since the SCD model is based on 3D FE simulations, it is computationally intensive and hence, expensive to complete a large number of realisations. One FE simulation takes approximately 30–40 min. For this reason, a response surface (RS) is trained in such a way that it yields the same model response $\theta_{calc}$ as the SCD model for the studied range of the input variables $X$. The response surface is a function (usually first or second order polynomial form) which approximate the physical or FE models but allow the reliability assessment of the investigated problem with resealable computational effort.

The design of experiment (DoE) procedure is used to explore the most significant combinations of the input variables $X$. Based on the developed FE simulation plan, the obtained outputs $\theta_{calc}$ are used to fit the response function.

Figure 1 summarises the methodology for the reliability analysis design for lateral cyclic loading. The framework starts with the uncertainty quantification from the available data (CPT, cyclic laboratory tests of the soil and metocean & aero-hydro-servo-elastic model) and the derivation of the stochastic input variables (soil stiffness, cyclic contour diagram, and storm event). The chosen stochastic variables are the input of the SCD model. Based on the stochastic input variables, a response surface is then trained to yield the same output (in terms of structural tilting) of the 3D FE simulations. The response surface is then used to calculate the probability of failure passing through the formulation of the limit state equation and the MC simulation. If the calculated probability of failure does not meet the target probability, then the foundation geometry has to be changed and the methodology is repeated to check whether the new design is safe.

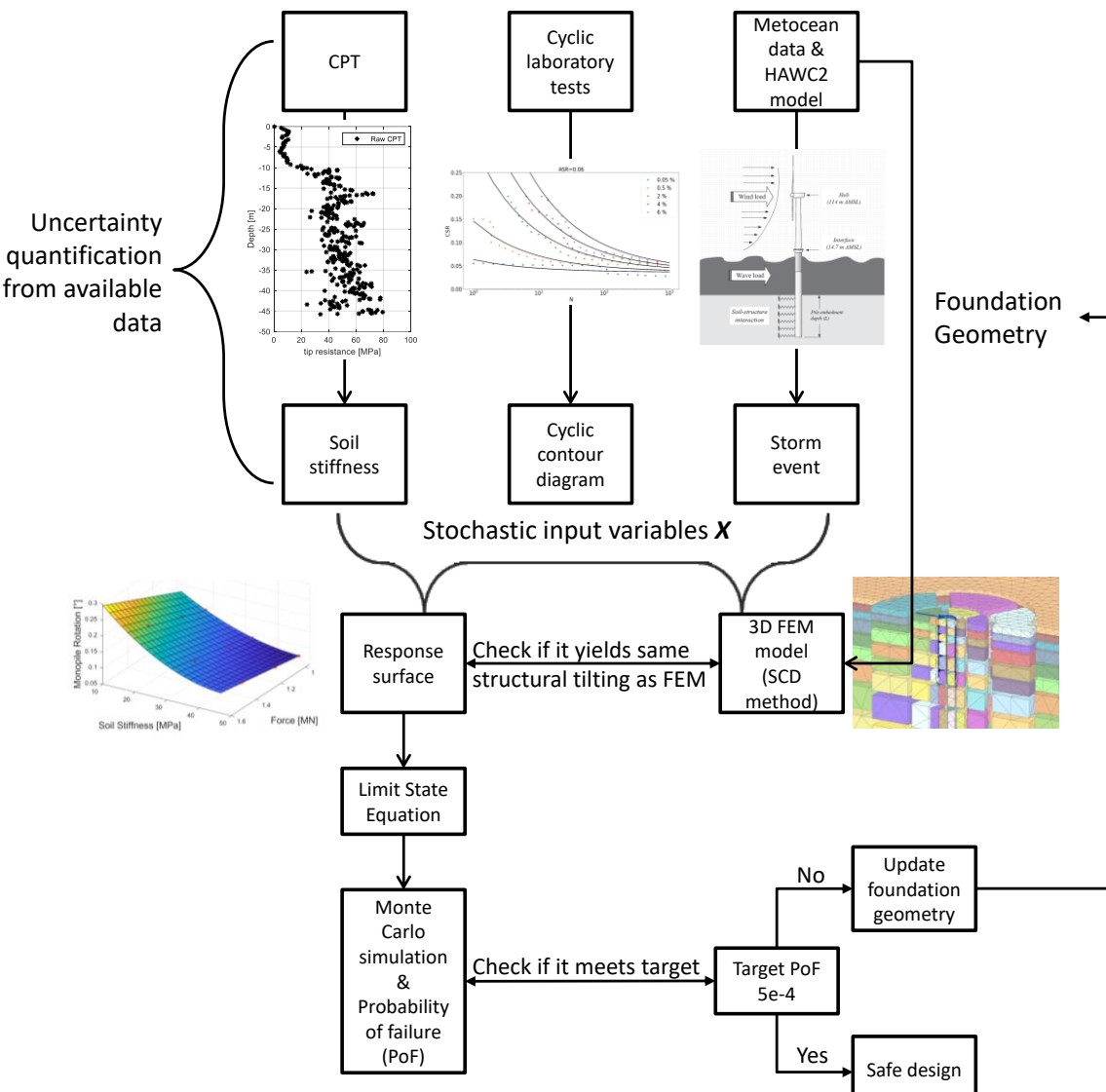

**Figure 1: Methodology of reliability analysis.**

## 3. Case study: Reliability design for a monopile supporting a 10MW wind turbine

In this section, firstly, the monopile pre-design (static load design step) is carried out in which the subsoil conditions of the
case study and the ULS design of the monopile geometry supporting a 10 MW wind turbine are explained. The pre-design of
the monopile is developed using the Hardening Soil model in finite element to predict the static response of the monopile.
Then the reliability framework for the cyclic load design shown in Figure 1 is applied to the monopile to check if the pre-
design satisfies the SLS criteria. The following subsections discuss the derivation of input uncertainties for the SCD method,
derivation of the response surface and probability of failure, and reliability index calculation.

**3.1 Monopile pre-design: subsoil condition and pile geometry**

For the present case study, a tip resistance from the cone penetration test (CPT) and the boring profile are used to determine the geotechnical properties and soil stratigraphy at the site, where the monopile is assumingly installed. A CPT is basically a steel cone which is pushed into the ground and the tip resistance is recorded. Base on the recorded tip resistance, soil
stratigraphy and soil properties can be empirically derived.

The CPT, shown in Figure 2, features an increase in the tip resistance with increasing depth, which is typical for sand. In combination with the borehole profile, the tip resistance from the CPT suggests that the soil can be divided into two different layers. At approximately –10 m there is a jump in the tip resistance marking a transition to another layer with a higher magnitude visible, leading to the conclusion that denser sand is present. The characterisation of the soil extracted from the
boreholes, shows the first layer (from 0 to –10 m) consisting of fine to medium sand and the second layer (from –10 m) of well-graded sand with fine gravel.

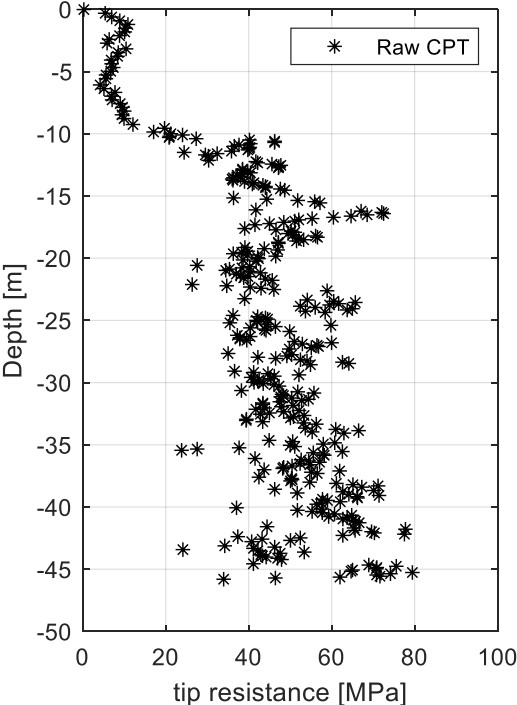

**Figure 2: CPT profile.**

To accurately predict the soil-structure-interaction and incorporate the rigid behaviour of the large diameter monopile, the ULS geotechnical verification of the preliminary design of the monopile is carried out, using the finite element method in PLAXIS 3D.

The monopile is modelled in PLAXIS as a hollow steel cylinder using plate elements. For the steel, a linear elastic material is assumed with a Young's modulus of 200 GPa and a Poisson coefficient of 0.3. The interface elements are used to account for the reduced shear strength at the pile's surface.

The soil model used is the Hardening Soil model with small-strain stiffness (HSsmall). The soil model parameters for the two layers are derived from the tip resistance (Figure 2) and listed in Table 1. The relative density (which is related to the soil porosity) of the two layers is calculated using the formula from Baldi et al. (Baldi et al., 1986) with the over-consolidated parameters (typical for offshore conditions) leading to a mean value of 70% and 90% for the first and second layers, respectively.

| Soil | Parameter | Value | Soil | Parameter | Value |
|---|---|---|---|---|---|
| Fine-medium sand | $E_{50}$ [MPa] | 33.3 | Medium-coarse sand | $E_{50}$ [MPa] | 98.3 |
| | $E_{oed}$ [MPa] | 33.3 | | $E_{oed}$ [MPa] | 98.3 |
| | $E_{ur}$ [MPa] | 99.9 | | $E_{ur}$ [MPa] | 295 |
| | m [-] | 0.5 | | m [-] | 0.5 |
| Depth: from 0 to -10 m | c [kN/m2] | 0.1 | Depth: from -10 m | c [kN/m2] | 0.1 |
| | $\varphi$ [°] | 39 | | $\varphi$ [°] | 42 |
| | $\psi$ [°] | 9 | | $\psi$ [°] | 12 |
| | $G_0$ [MPa] | 116 | | $G_0$ [MPa] | 196.6 |
| Relative density: 70% | $\gamma_{0.7}$ [−] | 0.0001 | Relative density: 90% | $\gamma_{0.7}$ [−] | 0.0001 |

**Table 1: Soil model parameters.**

The monopile design requires a loop between the structural and geotechnical engineers to update the soil stiffness and loads at the mudline level. A fully coupled aero-hydro-servo-elastic model using HAWC2 (Larsen and Hansen, 2015) is developed to perform the time-domain wind turbine load simulations (Velarde et al., 2019b). The soil structure interaction model is based on the Winkler-type approach, which features a series of uncoupled nonlinear soil springs (so called p-y curves) distributed at every 1 m. The force (p) – deformation (y) relations are extracted from the PLAXIS 3D model. At each meter section, the calculation of the force (p) is carried out by integrating the stresses along the loading direction over the surface. The displacement (y) is taken as the plate's displacement. The PISA project (Byrne et al. 2019) highlights that additional soil reaction curves components (distributed moment, horizontal base force and base moment) are needed in conjunction with the

195 p-y curves in order to have a more accurate soil structure interaction behaviour. For the present case study, the only use of the p-y curve extracted from FE is considered satisfactory.

The final pile design consists of an outer pile diameter at the mudline level of 8 m, a constant pile thickness of 0.11 m, and a pile embedment length of 29 m. The natural frequency of the monopile is 0.2 Hz and is designed to be within the soft-stiff region. Fatigue analysis of the designed monopile is also carried out (Velarde et al., 2019b). Figure 3.a shows the horizontal

displacement contour plot at 3.5 MN horizontal force, while Figure 3.b shows the horizontal load-rotation curve at the mudline.

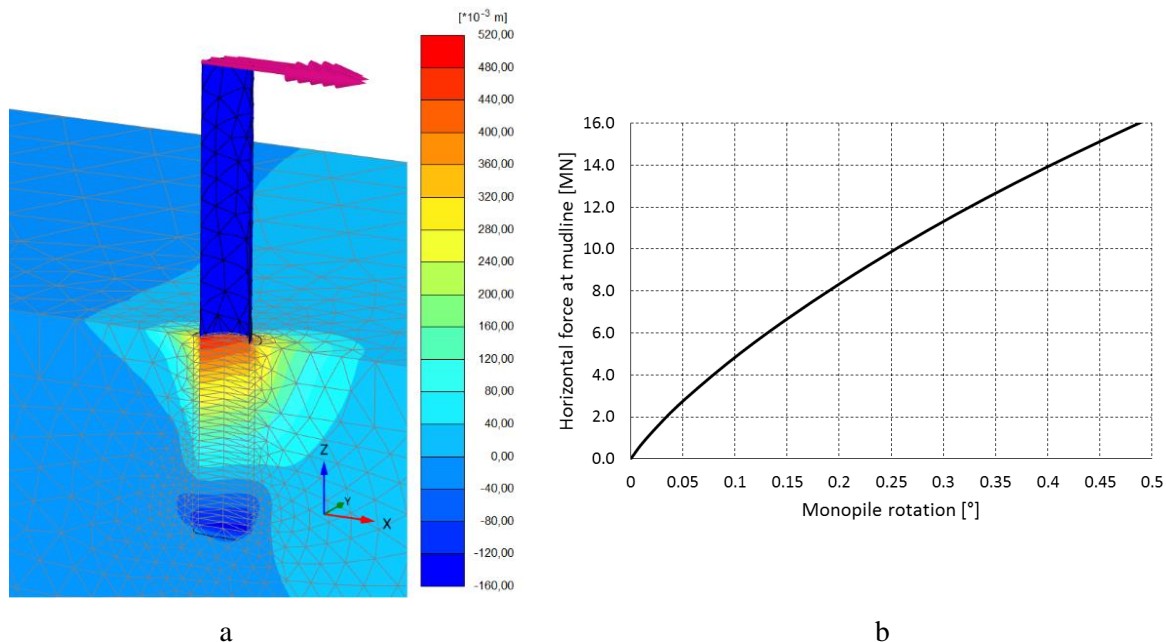

a                                    b

**Figure 3: (a) Horizontal displacement contour plot at 3.5 MN horizontal load; (b) Monopile rotation.**

### 3.2 Input uncertainties for the SCD model

The application of the SCD model requires three inputs: soil stiffness (for the Mohr-Coulomb soil model), cyclic contour

diagrams, and a design storm event. The laboratory testing and field measurements are used to estimate the inputs for the model. In this estimation process, different sources of uncertainty of unknown magnitude are introduced (Wu et al., 1989). These parameters then have to be modelled as stochastic variables with a certain statistical distribution.

### 3.2.1 Soil Stiffness

The uncertainties of the soil stiffness used in the SCD model is analysed. The soil model employed in the SCD method is the

210 Mohr-Coulomb model, with a stress-dependent stiffness (i.e., the stiffness increases with depth). For cyclic loading problems, the unloading-reloading Young's modulus $E_{ur}$ is used. This soil modulus is obtained from the tip resistance from the CPT test (Figure 2). The layering of the soil domain is assumed to be deterministic as explained in section 3.1.

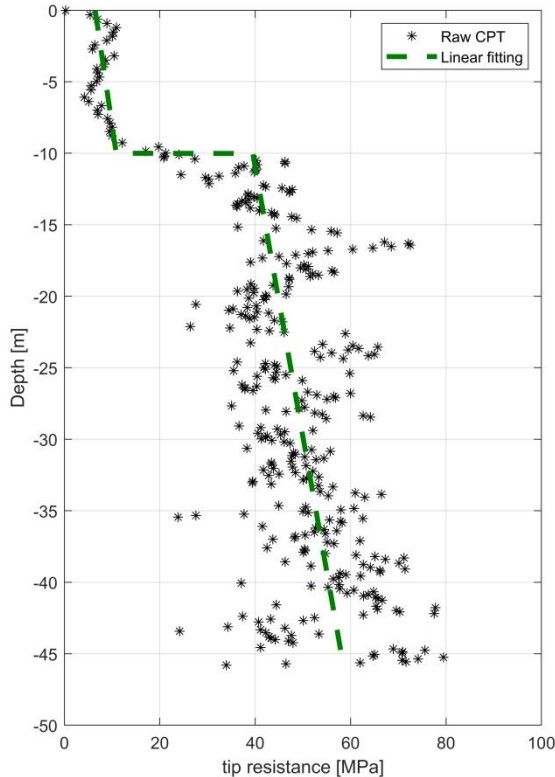

**Figure 4: Average tip resistance.**

The design tip resistance is established by means of the best-fit line in the data. A linear model is fitted to the data for each

layer (Figure 4, green line). The maximum likelihood estimation (MLE) is used for estimating the parameters of the linear

model along with the fitting error (assumed to be normally distributed and un-biased). From the MLE method, the standard

deviations and correlations of the estimated parameters (Sørensen, 2011) are obtained. The linear model is expressed by means

of Eq. (5) as below:

$q_c = X_a\, z + X_b + \varepsilon\,,$                                                                                      (5)

where $X_a$, $X_b$ are stochastic variables modelling parameter uncertainty related to the parameters a and b, respectively; $\varepsilon$ is the

fitting error; and $z$ is the depth (m). Table 2 shows a summary of the fitting parameters.

The residuals are then plotted to check the assumption of the normality of the model error. For the first layer (Figure 5.a), the

distribution of the residual is slightly skewed to the right. This means that the trend line underrepresents the tip resistance due

to the presence of high peeks at the boundary layer. For the second layer (Figure 5.b), a normal distribution about the zero

mean is visible, implying that a better fit is achieved.

| Parameter | Distribution | Mean | Standard Deviation |
|---|---|---|---|
| $X_a$ (1st layer) | Normal | -0.42 | 0.049 |
| $X_a$ (2st layer) | Normal | -0.53 | 0.024 |
| $X_b$ (1st layer) | Normal | 6.35 | 0.28 |
| $X_b$ (2st layer) | Normal | 34.05 | 0.72 |
| $\varepsilon$ (1st layer) | Normal | 0 | 3.14 |
| $\varepsilon$ (2st layer) | Normal | 0 | 16.06 [MPa] |
| $\rho_{X_a,X_b}$ (1st layer) | - | 0.86 | - |
| $\rho_{X_a,X_b}$ (2st layer) | - | 0.98 | - |

Table 2: Stochastic input variable for tip resistance.

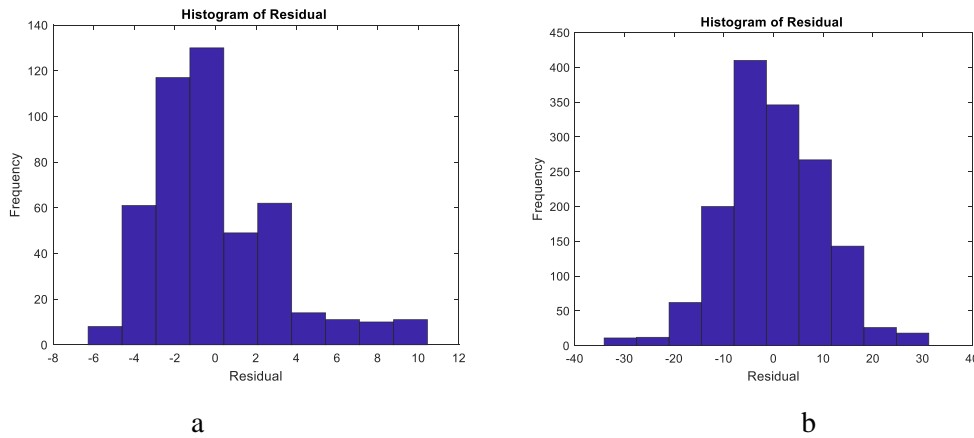

a                                         b

Figure 5: Histogram of residual for layer 1 (a) and layer 2 (b).

An empirical linear relationship is used to calculate the drained constraint modulus in unloading/reloading $E_s$ (Lunne et al., 1997, Lunne and Christoffersen, 1983):

$$E_s = X_\alpha \ q_c ,$$
(6)

where $X_\alpha$ is a unit-less stochastic variable. For over-consolidated sand, which is typical of offshore conditions, a value of $\alpha = 5$ is recommended (Lunne and Christoffersen, 1983). However, there is no unique relation between the stiffness modulus and the tip resistance because the $\alpha$ value is highly dependent on the soil, stress history, relative density, effective stress level, and other factors (Lunne et al., 1997, Bellotti et al. 1989, Jamiolkowski et al. 1988).

To understand the uncertainty in the stiffness modulus, α is treated as a stochastic normal variable varying from $\alpha_{min} = 3$ to $\alpha_{max} = 8$ with a mean μ = 5.5 and standard deviation σ = 1.25. The standard deviation is calculated by $(\alpha_{min} - \alpha_{max})/4$, assuming that 95.4% of the values are enclosed between the α values of 3 and 8.

Thus, the calculation of the drained constraint modulus in unloading/reloading, covering all possible uncertainties is summarised as follows:

$$E_s = \boldsymbol{X}_\alpha \left[ \boldsymbol{X}_a \, z + \boldsymbol{X}_b + \varepsilon \right], \tag{7}$$

Depending on the size of the foundation, the local fluctuation (physical uncertainty) of the tip resistance can have a significant impact on the structural behaviour. If the size of the foundation is large enough, the soil behaviour is governed by the average of the global variability of the tip resistance (mean trend value). For a smaller foundation, the local effect, i.e., the local physical variability of the tip resistance governs the soil behaviour. If the local variability of the tip resistance does not affect the foundation behaviour compared to the fitted linear model, it can be neglected. Moreover, the uncertainty related to the empirical formulation for calculating the soil stiffness ($\boldsymbol{X}_\alpha$), has a higher influence compared to the one used to approximate the tip resistance with a linear model ($\boldsymbol{X}_a, \boldsymbol{X}_b, \varepsilon$). The preliminary results show that the uncertainty associated with approximating the tip resistance with the mean trend line is negligible due to the size of the monopile. For this reason, $\boldsymbol{X}_a, \boldsymbol{X}_b, \varepsilon$ are considered deterministic at their mean value.

Figure 6 shows the variability of the soil modulus $E_s$ over depth. The red lines are the realisations, using the MC simulation by performing random sampling on the stochastic variable $\boldsymbol{X}_\alpha$. The black points are the deterministic multiplication of the tip resistance with a mean value of $\alpha = 5.5$.

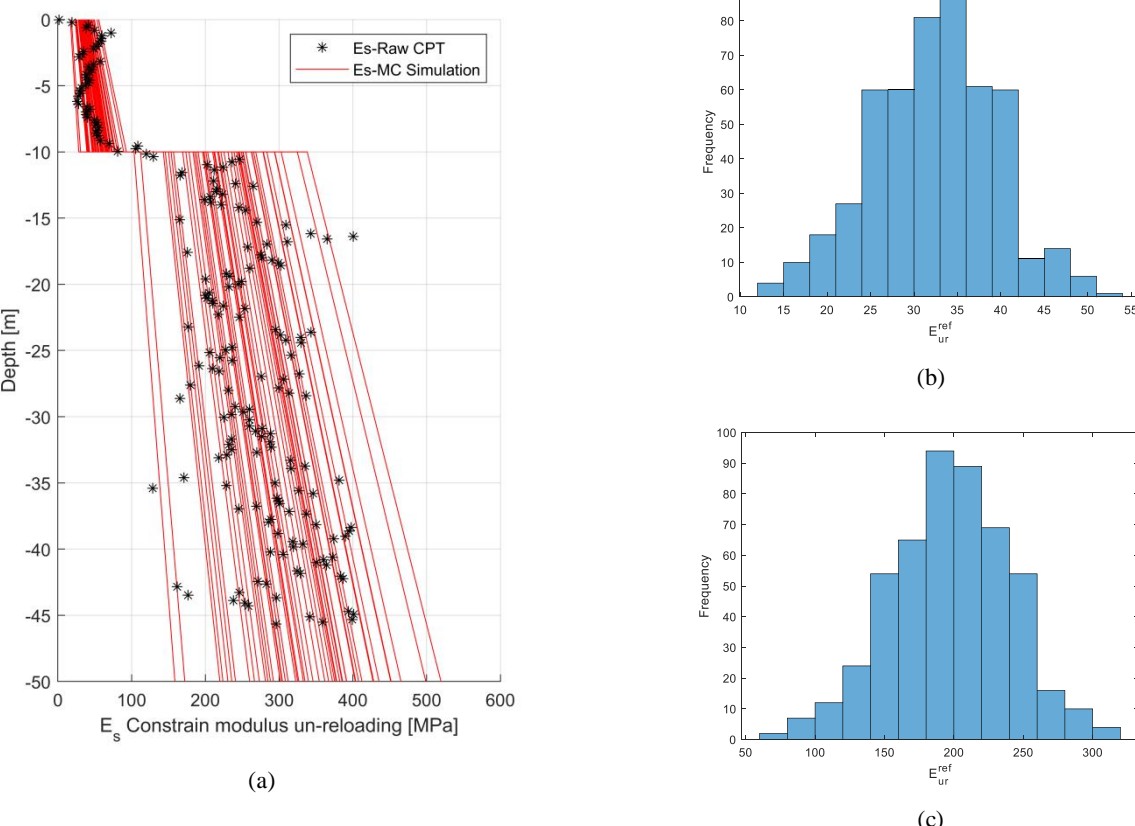

(a)

(b)

(c)

**Figure 6: (a) Variability of the soil modulus $E_s$ over depth; (b) Histogram of the soil stiffness at $z_{ref} = 0\ m$; (c) Histogram of the soil stiffness at $z_{ref} = -10\ m$;**

The drained constraint modulus in unloading/reloading $E_s$ is then converted to the drained triaxial Young's modulo in unloading/reloading $E_{ur}$ used in the Mohr–Coulomb soil model in PLAXIS. Assuming an elastic behaviour of the soil during unloading/reloading, $E_s$ and $E_{ur}$ can be related as:

$$E_{ur} = \frac{(1-v_{ur})}{(1+v_{ur})*(1-2*v_{ur})} E_s \ , \tag{8}$$

where $v_{ur}$ is the Poisson ratio (= 0.2)

The soil stiffness depends on the depth. In the Mohr–Coulomb model, a linear increase in the stiffness with depth is accounted for using the following formula:

$$E(z)_{ur} = E(z)_{ur}^{ref} + \left( z_{ref} - z \right) E_{inc} \ , \tag{9}$$

where $E(z)_{ur}$ is the Young's modulus for unloading/reloading at a depth $z$ ; $E(z)_{ur}^{ref}$ is the Young's modulo for unloading/reloading at a reference depth $z_{ref}$; and $E_{inc}$ is the increment of the Young's modulus. Using this equation for a given input value of $E_{ur}^{ref}$ and the increment $E_{inc}$, $E_{ur}$ can be derived at a specific depth below the surface and compared to

$E_s$, as specified in the design soil profile. For all realisations of different soil stiffnesses (Figure 6 red lines), $E_{ur}^{ref}$ and the increment $E_{inc}$ are calculated:

- For the first layer at $z_{ref} = 0$ (Figure 6.b): $\mu_{E_{ur}^{ref}} = 32.25\ MPa$ and $\sigma_{E_{ur}^{ref}} = 7.06\ MPa$

- For the second layer at $z_{ref} = -10\ m$ (Figure 6.c): $\mu_{E_{ur}^{ref}} = 196.90\ MPa$ and $\sigma_{E_{ur}^{ref}} = 43.14\ MPa$

Other soil properties, such as specific weight, friction angle, and relative density are considered to be deterministic. A full positive correlation between the two soil layer stiffness is assumed.

### 3.2.2 Cyclic contour diagrams

The aim of the contour diagrams is to provide a 3D variation of the accumulated permanent strain in the average stress ratio (ASR), which is the ratio of the average shear stress to the initial vertical pressure or confining pressure, the cyclic stress ratio (CSR), which is the ratio of the cyclic shear stress to the initial vertical pressure or confining pressure, and the number of cycles (N). An extensive laboratory test campaign is needed to have an accurate 3D contour diagram. The laboratory campaign generally consists of carrying out different regular cyclic load tests with different average and cyclic amplitude stresses for a certain number of cycles.

For this work, a series of undrained (constant volume) single-stage two-way cyclic simple shear tests were performed at the Soil Mechanics Laboratories of the Technical University of Berlin. The tests were carried out on reconstituted soil samples. The samples were prepared by means of air pluviation method. The initial vertical pressure was 200 kPa and no pre-shearing was considered.

The cyclic behaviour of the upper layer of sand was evaluated with samples prepared at a relative density of 70%. For the lower layer sand, a 90% relative density was used. Two-way cyclic loading tests were carried out, testing different combinations of ASR and CSR. All the tests were stopped at 1000 cycles or at the start of the cyclic mobility phase. For the results on the cyclic behaviour of various tests and relative densities, refer to Zorzi et al. (Zorzi et al., 2019.b).

All the data extracted from the laboratory tests were assembled in a 3D matrix (ASR, CSR, N) and a 3D interpolation of the permanent shear strain ($\gamma_p$) was created to map the entire 3D space. The repeatability of the cyclic simple shear tests is an important aspect to consider in evaluating the uncertainties in the cyclic contour diagram. Cyclic simple shear tests feature a low repeatability for dense sand, which can be attributed to the relatively small specimen size used for testing (Vanden Bergen, 2001). This makes the cyclic tests sensitive to sample preparation, this resulting in, for example, different initially measured relative densities, soil fabric, and void ratio non-uniformities.

Owing to this variability of the test, a mathematical formulation was fitted to the raw interpolation. For this reason, different two-dimensional slices (CSR vs. N) at different ASR values were extracted. Figure 7 represents a slice of ASR equal to 0.06. The different coloured points represent the strain surfaces $\gamma_p$ for different levels of deformation. The raw interpolation of data and the uncertainty related to the low sample repeatability of the tests cause an unrealistic non-smooth shape of the strain

surfaces. Therefore, each slice is assumed to follow a power law function (variation of CSR as power of N) for different strain
levels and then calibrated to fit the data. Finally, the calibrated strain surfaces are interpolated to create the final smooth 3D
contour diagram. This procedure and its validation are explained in Zorzi et al. (Zorzi et al., 2019a).

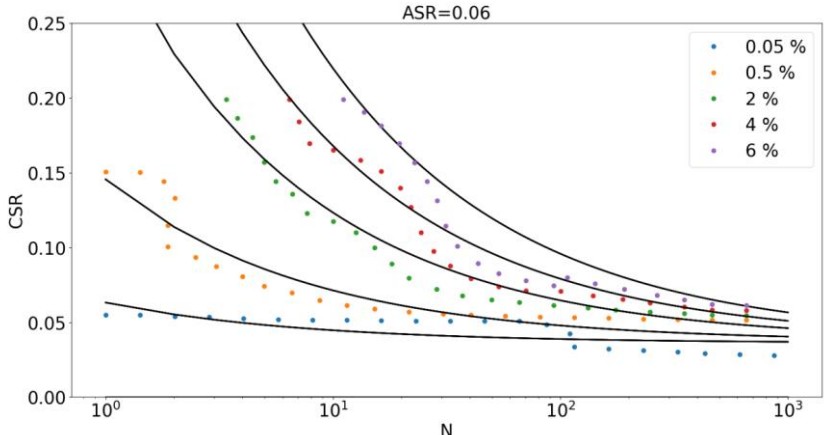

**Figure 7: Slice of the cyclic contour diagram.**

The power law function can be written in the form of Equation (1).

$$CSR = \mathbf{X}_c\, N^{X_d} + \mathbf{X}_e + \varepsilon\,,\qquad\qquad\qquad (10)$$

where $\mathbf{X}_d$ represents the shape of the curve; $\mathbf{X}_c$ is a scaling factor; $\mathbf{X}_e$ is the intersection with the CSR axis, and $\varepsilon$ is the fitting
error. Using the maximum likelihood method (MLM) it is possible to fit the mathematical model and estimate the standard
deviation of the fitting error and the standard deviation of the parameters $c$ and $e$. During the fitting procedure, the shape
parameter $d$ is assumed fixed at –0.35 for the lower layer and –0.50 for the upper layer.

Based on the results of the fitting procedure, a standard deviation of the fitting error of 0.008 is chosen for the two diagrams
for the two soils. The parameters c and e are considered deterministic, as the standard deviation associated is very low.
Preliminary simulations show that the uncertainty of $a$ and $c$ derived from the MLM has less influence than the uncertainty in
the fitting error.

It has to be noted that the fitting error, to some extent, reflects the uncertainties of repeatability of the tests. Moreover, the
relative density of the soil samples is based on the empirical relation applied to the tip resistance (section 3.1). To account for
the uncertainty in the relative density, different sets of contour diagrams should have been derived from several tests performed
with soil samples at different relative densities.

The contour diagrams for two different ASR slices are presented in Figures 8 and 9 for the upper and lower layers, respectively.

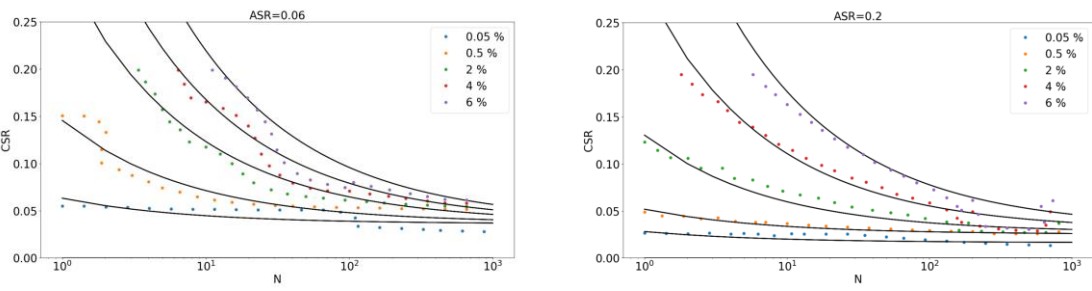

**Figure 8: Cyclic contour diagram for the first layer.**

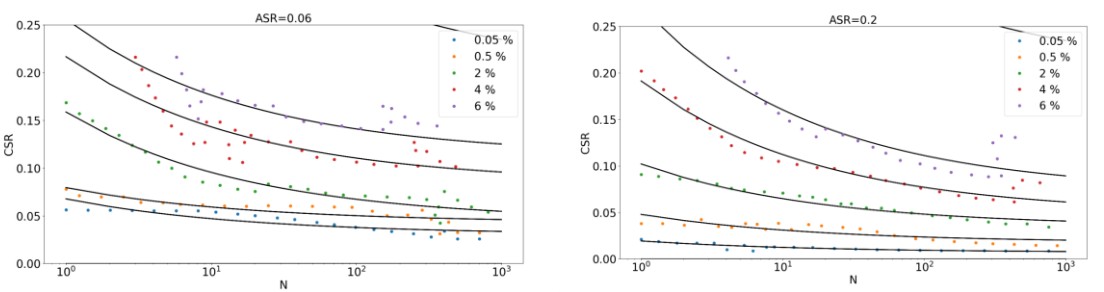

**Figure 9: Cyclic contour diagram for the second layer.**

### 3.2.3 Load uncertainty

The load input parameter for the SCD model is characterised by a regular loading package with a mean and cyclic amplitude load, and an equivalent number of cycles (hereafter, called "load inputs" for simplicity). In common practice, the structural engineer provides the irregular history at mudline level by means of the aero-hydro-servo-elastic model. Therefore, a procedure is needed to transform the irregular design storm event to one single regular loading parcel. The environmental load used for the cyclic loading design relies on the chosen return period for the load. The statistical distribution of the environmental loads is then based on different return periods.

The design storm event is here defined as a six hour duration of the extreme load (also called the peak of the storm) (DNV-GL, 2017). The underlying assumption in considering only the peak is that most of the deformations, which the soil experiences, happen at the peak of the storm. The considered design load case is DLC 6.1 (IEC, 2009; BSH, 2015), i.e., when the wind turbine is parked and yaw is out of the wind. The ULS loads are considered for the cyclic load design.

To derive the irregular load history at the mudline level, the fully coupled aero-hydro-servo-elastic model is developed in the wind turbine simulation tool, HAWC2. Based on 5-year in-situ metocean data from the North Sea, the environmental contours for different return periods are derived as shown in Figure 10 (Velarde et al, 2019a). The marginal extreme wind distribution is derived using the peak-over-the-threshold method for wind speed above 25 m/s. Furthermore, it is assumed that maximum responses are given by the maximum mean wind speed and conditional wave height for each return period (red point in Figure 10).

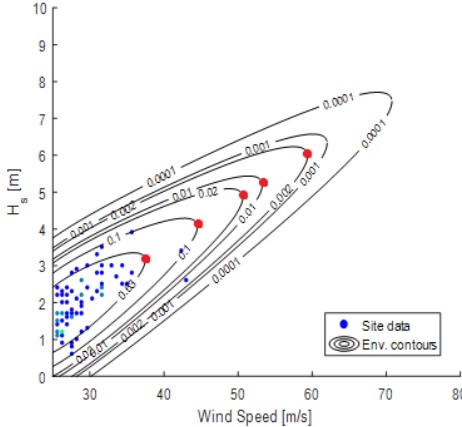

**Figure 10: Environmental contour plot for extreme sea states (Velarde et al, 2019a).**

The five design sea states for maximum wind speed are summarised in Table 3. To account for short-term variability in the responses, 16 independent realisations are considered for each design sea state.

| Annual Exceedance Probability (q) | Return period [year] | Wind speed Uw [m/s] | Wave height Hs [m] | Wave period Tp [s] |
|---|---|---|---|---|
| 0.63 | 1 | 37.4 | 3.17 | 7.95 |
| 0.10 | 10 | 44.5 | 4.10 | 8.84 |
| 0.02 | 50 | 50.6 | 4.90 | 9.54 |
| 0.01 | 100 | 53.3 | 5.24 | 9.83 |
| 0.002 | 500 | 59.4 | 6.04 | 10.44 |

**Table 3: Design sea state for maximum wind speed.**

Time-domain simulations provide an irregular force history of 10 min at the mudline. To transform the 10 min irregular loading to a 6-h storm, each 10-min interval is repeated 36 times.

The irregular load histories have to be simplified to one equivalent regular package with a specific mean and cyclic load
 amplitude and an equivalent number of cycles that lead to the same damage accumulation (accumulation of soil deformation) as that of the irregular load series.

The following procedure is used (Andersen, 2015):

- The rainflow counting method is utilised to break down the irregular history into a set of regular packages with different combination of mean force $F_a$ and cyclic amplitude force $F_{cly}$ and number of cycles $N$. Figure 11 shows an
 example of the output from the rainflow counting.

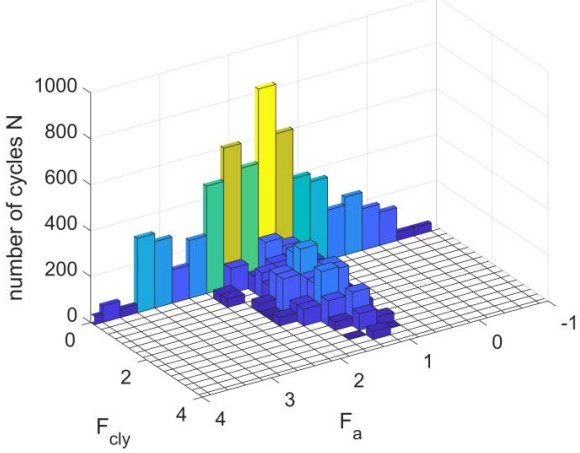

**Figure 11: Rainflow matrix for 100 year return period wind speed.**

- All the bins are ordered with increasing maximum force $F_{max}$ obtained from the sum of the mean and the cyclic amplitude ($F_{max} = F_a + F_{cly}$).

- 3D contour diagrams in conjunction with the strain accumulation method [21, 2] are then used to calculate the accumulation of deformation. After scaling the loads to shear stresses, the result of this procedure gives the equivalent number of cycles for the highest maximum force $F_{max}^{max}$, which in turn gives the same accumulation of deformation of the irregular load history.

This procedure is applied for all simulations with different return periods.

To obtain a statistical distribution, the mean force, cyclic amplitude force, and the equivalent number of cycles are plotted versus the probability of non-exceedance for each return period.

The black points in the three following figures are, respectively, the mean load, cyclic amplitude and number of cycles of the regular packages obtained from the previous procedure and plotted vs. the probability of not exceedance for each return period. Assuming that for each return period the black points have a normal distribution, the 0.50 fractile (red circles) and the 0.95

fractile (blue circles) are obtained.

The statistical distributions for the loads are derived by fitting a Gumbel distribution to the 0.95 fractile values (NORSOK, 2007). The MLM is employed to fit the cumulative Gumbel distribution to the extreme response (blue circles). The cumulative density function distribution is defined as:

$$CDF(x) = \exp(-\exp(-(x-\alpha)/\beta)) , \qquad (11)$$

$$\mu = \alpha + \beta\, 0.5772 , \qquad (12)$$

$$\sigma = \frac{\pi}{2.44}\, \beta , \qquad (13)$$

The Table 4 summarises the parameters of distribution for the three load inputs. The standard deviation of the fitting error is small, marking a good fitting of the distribution function.

| Load | $\alpha$ | $\beta$ | $\mu$ | $\sigma$ | $\sigma_\varepsilon$ |
|------|----------|---------|-------|----------|----------------------|
| $F_a$ | 1.092 | 0,113 | 1.158 [MN] | 0.382 [MN] | 0.0040 |
| $F_{cly}$ | 3.66 | 0.093 | 3.71 [MN] | 0.347 [MN] | 0.011 |
| $N_{eq}$ | 329.75 | 70.08 | 370.2 [Cycles] | 9.49 [Cycles] | 0.024 |

**Table 4: Gumbel parameters of the distribution for the load inputs.**

Looking at the distribution of Figure 12 a-b-c, larger 0.50 fractiles (red circles) are present when increasing the return period. This is more pronounced for the mean force and is expected because the higher the return period, the higher is the mean pressure on the wind turbine tower.

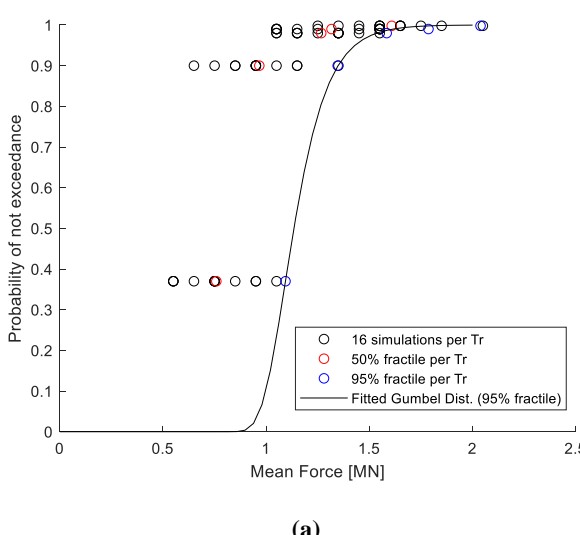

**(a)**

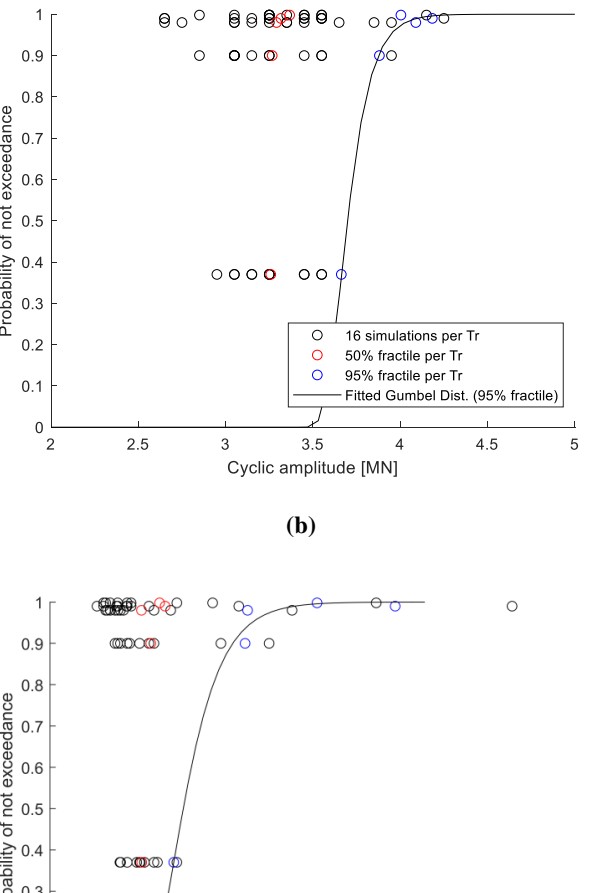

**(b)**

**(c)**

**Figure 12: Distribution of the load inputs.**

The scatter for each return period is more significant when the return period increases. This can have different reasons, for example, the "rare" storms with a lower probability of occurrence could give more non-linearity problems varying the wave and wind speeds in the aero-hydro-servo-elastic model. It could also depend on the model uncertainty in the time domain simulations.

The correlation coefficients $\rho$ for the 0.95 fractile values between the mean and cyclic loads and the equivalent number of cycles are: $\rho_{F_a-F_{cly}} = 0.77$, $\rho_{N_{eq}-F_{cly}} = 0.81$, and $\rho_{N_{eq}-F_a} = 0.85$. The three coefficients mark a strong positive correlation between the three load inputs.

### 3.2.4 Model error

This type of error is difficult to estimate because it requires the validation of the numerical error against different model tests.
In the case of the SCD model, this error is arising due to the simplification of the model for a much more complex behaviour of the soil-structure-interaction under cyclic loading. The model error $\varepsilon_{model}$ is estimated as a random variable and multiplied to predict the structural tilting (Eq. 14). The model error is assumed to be normally distributed with a unitary mean and a coefficient of variation of 10%. Ideally, this model uncertainty should be quantified comparing the results from the SCD model with several different test results. However, such a large number of tests is not feasible.

$$g(X) = \theta_{max} - \varepsilon_{model}\,\theta_{calc}(X)\,, \tag{14}$$

### 3.3 Derivation of the response surface

The stochastic variables are summarized in Table 5. For simplicity, a full correlation between the soil stiffness of the two layers and the loads is assumed.

| $X$ | Unit | PDF | $\mu$ | $\sigma$ | CoV [%] | $\rho$ |
|---|---|---|---|---|---|---|
| Soil stiffness Layer 1 $E_{ur}^{ref}$ | MPa | Normal | 32.25 | 7.06 | 21.9 | |
| Soil stiffness Layer 2 $E_{ur}^{ref}$ | MPa | Normal | 196.90 | 43.14 | 21.9 | 1 |
| Cyclic contour diagrams fitting error $CCD_{err}$ | - | Normal | 0 | 0.008 | - | - |
| Input load $F_a$ | MN | Gumbel | 1.158 | 0.382 | 32.9 | |
| Input load $F_{cly}$ | MN | Gumbel | 3.71 | 0.347 | 9.3 | 1 |
| Input load $N_{eq}$ | Cycles | Gumbel | 370.2 | 9.49 | 2.5 | |

**Table 5: Summary of the stochastic variables.**

Once the stochastic variables are defined, the 3D FEM model has to be substituted by a response surface.

The DoE is used to obtain the training point from the FE simulation. As most of the variables are correlated, three stochastic variables are considered: (i) the stiffness of the upper soil layer $E_{ur}$, (ii) the fitting error of the cyclic contour diagram $CCD_{err}$, and (iii) the mean load $F_a$. The independent input stochastic variables have the statistical distribution shown in Table 5. For
each factor, three different levels are assumed: minimum value $\mu - 2*\sigma$, average value $\mu$ and maximum value $\mu + 2*\sigma$. A full factorial design in three levels is implemented. Therefore, a total of 33 simulations are needed to explore all possible combinations.

Based on visual inspection of the output from the 3D FEM model, a second-order polynomial function is fitted to the sample data. The linear regression method is used to estimate regression coefficients of the polynomial function. The following function is the outcome of the linear regression analysis:

$$\theta_{calc} = 0.248\, F_a - 0.007\, E_{ur}\, F_a - 0.144\, F_a\, CCD_{err} + 0.0000746\, E_{ur}{}^2\, F_a + \varepsilon_{fit}\,, \tag{15}$$

An un-biased fitting error ($\varepsilon$) with normal distribution is assumed and the estimate of residual standard deviation ($\sigma_{\varepsilon_{fit}}$) is 0.0013. R-squared is a statistical measure of how close the data is to the fitted regression line. For the fitted function, the R-squared value is 0.9984 underlining a good fit of the function to the data and hence the choice of the initial choice of the second-order polynomial function.

Figure 13.a shows the function at the $CCD_{err} = 0$ (the mean value). The surface shows that at a lower soil stiffness and a high force, a higher rotation of the monopile is reached. Values higher than 0.25 are considered as failures. The red points are from the numerical simulations. The 3D plot (Figure 13.b) shows the response surface for the mean value of the force $F_a = 1.158$ MN. It is apparent that the fitting error for the contour diagram is small and thus, does not have a significant influence on the results.

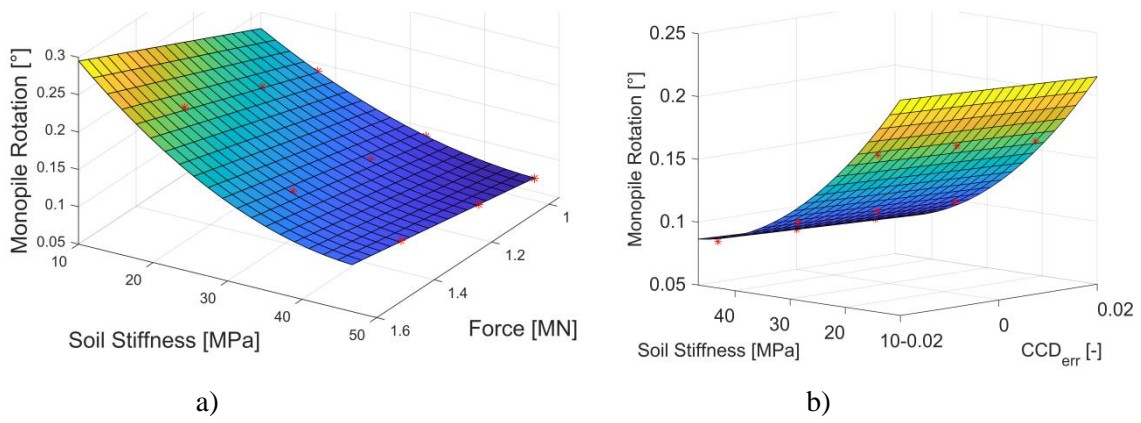

a)                                             b)

**Figure 13: Response surfaces.**

**3.4 Reliability analysis**

The limit state function is written as:

$$g(X) = 0.25° - (0.248\, F_a - 0.007\, E_{ur}\, F_a - 0.144\, F_a\, CCD_{err} + 0.0000746\, E_{ur}{}^2\, F_a + \varepsilon_{fit})\, \varepsilon_{model}\,, \tag{16}$$

$10^7$ MC simulations were performed by random sampling of the input stochastic variables. This number was the minimum required to keep the relative error of the reliability index lower than 1%. The stochastic variables and their probability distribution functions are given in Table 5. The derivation of the design mean, and standard deviation are explained in section 3.2.

With the analysed monopile design, the annual probability of failure is 2.7000e-05 and the corresponding annual reliability index is 4.03. This means, that the monopile meets the target reliability index of 2.9–3.3 and is considered safe for long-term behaviour in terms of rotation accumulation for the design storm event.

### 3.5 Sensitivity analysis

The sensitivity analysis of the stochastic input variables on the reliability index is conducted by varying the coefficient of variation one at a time for each input (0.5 CoV and 2 CoV). The inclination of dashed lines in Figure 14 marks the sensitivity of the stochastic variable. Mean force $F_a$ and the soil stiffness $E_{ur}$ are both influencing the reliability index significantly more than the fitting error and numerical model error do.

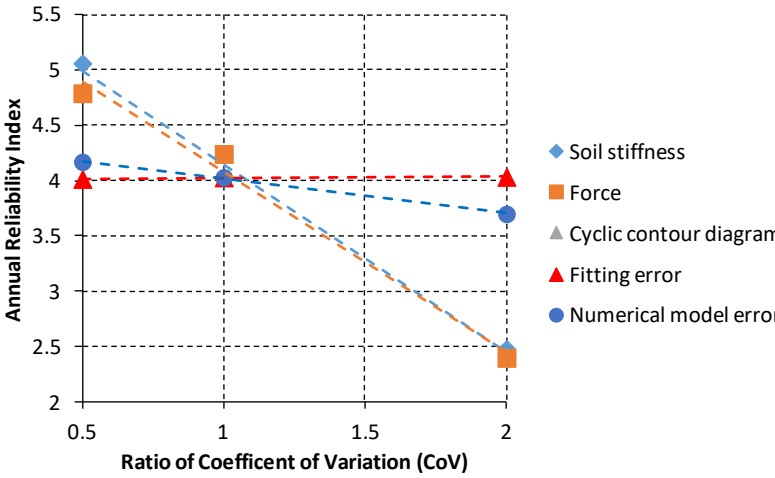

**Figure 14: Sensitivity plot.**

### 4. Conclusion

During the lifetime of wind turbines, storms, typhoons, or seismic action are likely to cause permanent deformation of the structure owing to the accumulation of plastic strain in the soil surrounding the foundation. The serviceability limit state criteria require that the long-term structural tilting does not exceed the operational tolerance prescribed by the wind turbine

manufacturer (usually less than 1°) with a specific target reliability level. In this study, the SLS design for long-term structural tilting is addressed within a reliability framework. This framework is developed based on the 3D FE models for the prediction of the SSI under cyclic loading. For the case study of a large monopile installed on a typical North Sea environment, a reliability index of 4.03 was obtained. Sensitivity analysis also shows that uncertainties related to the soil stiffness and the environmental loads significantly affect the reliability of the structure. For regions where assessment against accidental loads due to typhoons

are necessary, uncertainty of the extreme environmental loads can increase by up to 80%. Such load scenarios can significantly reduce the reliability index, and therefore become the governing limit state.

A discussion has to be started in the offshore community regarding the very strict tilting requirement (i.e., 0.25°). This very small operational restriction can lead to foundations of excessively large dimensions, which are unfeasible from an economic point of view. On the other hand, a less strict verticality requirement (which could be a function of the dimension and type of the installed wind turbine), for example an angle of rotation of 1–3°, could lead to a smaller foundation size and still meet the safety requirements. For this reason, by means of aero-elastic analyses, the investigation of the position of the natural frequency of the whole system and fatigue analysis should be carried out when a wind turbine is tilted at 1–3°. Allowing a less stringent tilting of the foundation can also be beneficial during the monopile installation. A small foundation dimension saves vessel and equipment cost, which contributes significantly to the overall cost reduction of the foundation.

In this paper, a simplified model to calculate the permanent rotation (the SCD method) is implemented. It is noted that other models of varying complexity can also be used in the proposed probabilistic framework. If new inputs are introduced, the respective uncertainties should be considered in the reliability calculation and the function for the response surface should be adjusted accordingly.

### Code and data availability

The codes can be made available by contacting the corresponding author.

### Author contribution

Zorzi, Mankar, Velarde and Sørensen designed the proposed methodology. Zorzi prepared the manuscript with the contributions from all co-authors.

### Competing interests

The authors declare that they have no conflict of interest.

### Acknowledgement

This research is part of the Innovation and Networking for Fatigue and Reliability Analysis of Structures - Training for Assessment of Risk (INFRASTAR) project. This project has received funding from the European Union's Horizon 2020 research and innovation program under the Marie Skłodowska-Curie grant agreement No. 676139. The laboratory tests are provided by the Chair of Soil Mechanics and Geotechnical Engineering of the Technical University of Berlin. The authors are grateful for the kind permission to use those test results.

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
