# Peer review of "Reliability analysis of offshore wind turbine foundations under lateral cyclic loading"

_Wind Energy Science, 2019_

## Referee Comment (RC1) · Federico Pisano (Referee) · 18 Sep 2019

**GENERAL COMMENTS**

This paper concerns an important issue in the design of support structures for offshore wind turbines, namely the estimation of the permanent tilt suffered by monopile foundations under long-lasting cyclic loading. Specifically, the Authors propose a probabilistic framework to obtain a reliability index (or probability of failure) associated to a specific monopile design - when driven by SLS requirements. This work appears sufficiently relevant and cross-disciplinary, the combination of reliability analysis and (simplified) geotechnical analysis of monopile cyclic tilt takes a step forward with respect to the current (and mostly deterministic) state of practice. I would, however, recommend revision

at several levels. The paper seems to be mainly written for a geotechnical readership, something probably not in the spirit of the WES journal. Given the many sources of existing knowledge contributing to the proposed analysis framework, it would be useful (and certainly fair) to acknowledge more extensively previous research.

SPECIFIC COMMENTS

- very geotechnical terminology is widely used throughout the text – concepts like "undrained conditions", "CPT", "relative density" are not necessarily in the background of many WES readers. Two possible solutions: (1) rephrasing and explaining "geo-jargon" (terms like porosity, constant-volume conditions, etc. will make more "common-sense" than those currently adopted); (2) creating an appendix in which important geotechnical concepts are briefly explained;

- in the introduction, the design of offshore foundations is described as dominated by ULS and/or SLS requirements. The important role played by fatigue limit states (FLS) should be clearly acknowledged and integrated in the global picture;

- the final reference list is very limited. While the main contribution of the Authors lies in the proposed probabilistic framework, existing geotechnical knowledge is instrumentally used. More references to existing literature should be introduced in relation to (i) lateral response of offshore monopiles under cyclic loading, (ii) probabilistic analysis in foundation engineering, (iii) representation of cyclic soil behaviour through the notion of "cyclic contour diagrams", (iv) theory underlying the derivation of the "response surface".

- the "soil cluster degradation" (SCD) method adopted by the Authors to predict cyclic monopile tilt sounds related to the work of Achmus and co-workers in Hannover. Please acknowledge any connections to such thread of research, if any;

- it is appreciated that the paper focuses on the development and use of the probabilistic framework. There seems to be, however, little concern about the validation of

its single "building stones". In particular, the abovementioned SCD method relies on a number of simplifying assumptions, some of which would be found quite "crude" after closer inspection. It would be good to remark the generality of the proposed framework, and suggest the inclusion of more accurate geotechnical analysis as new/better methods become available in the near future;

- in section 3.1 monopile design is "based on the Winkler-type approach", with p-y curves used exclusively. However, the PISA project has clearly put in evidence that more components of soil reactions are needed to properly analyse the performance of stubby monopiles (i.e., with low L/D ratio). Limitations in this respect should also be openly mentioned;

- preliminary 3D FE simulations seem to have been performed by using PLAXIS' Hardening Soil model. Later in the text reference to the simpler Mohr-Coulomb model is made. Please explain this seeming mismatch;

- it could be helpful to complement Fig. 6 with an explicit representation of the Young's modulus probability distribution at one or more selected depths.

TECHNICAL CORRECTIONS

I am not suggesting specific text amendments, but I feel that letting a native English speaker review the final manuscript would be beneficial.

---

## Referee Comment (RC2) · Anonymous Referee #2 · 27 Nov 2019

This paper presents a methodology for reliability analysis of complex structures, and applies it to offshore wind turbines. The paper is well written for most part and presents relevant information. I have the following suggestions for improving the manuscript: -Reference or background information for the chosen stochastic input variable parameters would be needed. -Justification for choosing the 2nd-order polynomial response function would be helpful. -The discussion about target reliability index on pages 3 and 4 needs changes. How can it be argued that SLS target should be in the range given, when Eurocode specifies it as 2.9? -The reliability analysis results presented in Section 3.4 need to be improved, some discussion on the design values of the random parameters would be needed. -The paper contains a number of typos.

---

## Author Comment (AC1) · 27 Apr 2020

The authors are very grateful for the comments and suggestions which have enhanced the quality of the paper. The replies are below in red.

GENERAL COMMENTS

This paper concerns an important issue in the design of support structures for offshore wind turbines, namely the estimation of the permanent tilt suffered by monopile foundations under long-lasting cyclic loading. Specifically, the Authors propose a probabilistic framework to obtain a reliability index (or probability of failure) associated to a specific monopile design - when driven by SLS requirements. This work appears sufficiently relevant and cross-disciplinary, the combination of reliability analysis and (simplified)geotechnical analysis of monopile cyclic tilt takes a step forward with respect to the cur-rent (and mostly deterministic) state of practice. I would, however, recommend revision at several levels. The paper seems to be mainly written for a geotechnical readership, something probably not in the spirit of the WES journal. Given the many sources of existing knowledge contributing to the proposed analysis framework, it would be useful (and certainly fair) to acknowledge more extensively previous research.

As you pointed out, this paper deals with a large range of disciplines with a special focus on geotechnics. The authors have tried to simplify as much as possible some concepts in order to make it clearer also for non-geotechnical readerships, but it is in our hope that the WES journal will also expand to the geotechnical side of the wind energy being as well a very important aspect when designing the offshore foundations. We have improved the paper at several levels and have replied to your comments.

SPECIFIC COMMENTS

- very geotechnical terminology is widely used throughout the text – concepts like "undrained conditions", "CPT", "relative density" are not necessarily in the background of many WES readers. Two possible solutions: (1) rephrasing and explaining "geojargon" (terms like porosity, constant-volume conditions, etc. will make more "commonsense" than those currently adopted); (2) creating an appendix in which important geotechnical concepts are briefly explained;

- The geotechnical terminologies are simplified and explanations of some concepts like CPT or cyclic contour diagrams are given.

- in the introduction, the design of offshore foundations is described as dominated by ULS and/or SLS requirements. The important role played by fatigue limit states (FLS)should be clearly acknowledged and integrated in the global picture;

- The global picture for the design stage is introduced. The major division in static load design and cyclic load design is explained. For the cyclic load design, the ULS, SLS and FLS limit state are briefly explained.

- the final reference list is very limited. While the main contribution of the Authors lies in the proposed probabilistic framework, existing geotechnical knowledge is instrumentally used. More references to existing literature should be introduced in relation to (i)lateral response of offshore monopiles under cyclic loading, (ii) probabilistic analysis in foundation engineering, (iii) representation of cyclic soil behaviour through the notion of "cyclic contour diagrams", (iv) theory underlying the derivation of the "response surface".

- The introduction section has been modified including different methods used in current practice for cyclic load design, probabilistic methods in offshore foundation engineering and regarding the theory behind the cyclic contour diagrams.

- the "soil cluster degradation" (SCD) method adopted by the Authors to predict cyclic monopile tilt sounds related to the work of Achmus and co-workers in Hannover. Please acknowledge any connections to such thread of research, if any;

- The work of Achmus and colleagues is added as a reference.

- it is appreciated that the paper focuses on the development and use of the probabilistic framework. There seems to be, however, little concern about the validation of its single "building stones". In particular, the abovementioned SCD method relies on a number of simplifying assumptions, some of which would be found quite "crude" after closer inspection. It would be good to remark the generality of the proposed framework, and suggest the inclusion of more accurate geotechnical analysis as new/better methods become available in the near future;

- This is a very good remark because the proposed framework can be used for more complicated geotechnical analysis in order to calculate the permanent rotation. However, more complex models may require more inputs and hence a more complex calibration of the surface response. This remark is added in the conclusion.

- in section 3.1 monopile design is "based on the Winkler-type approach", with p-y curves used exclusively. However, the PISA project has clearly put in evidence that more components of soil reactions are needed to properly analyses the performance of stubby monopiles (i.e., with low L/D ratio). Limitations in this respect should also be openly mentioned;

- This limitation is now mentioned in the paper (section 3.1). The focus was on the cyclic behavior, hence the authors thought that the modeling of just the p-y curves extracted from the FE model were accurate enough.

- preliminary 3D FE simulations seem to have been performed by using PLAXIS' Hardening Soil model. Later in the text reference to the simpler Mohr-Coulomb model is made. Please explain this seeming mismatch;

- The name of the section 3.1 is changed in order to avoid confusion. The HS model is used in the pre-design of the monopile (static load design). While the Mohr-Coulomb is used in the SCD method for the cyclic load design.

- it could be helpful to complement Fig. 6 with an explicit representation of the Young'smodulus probability distribution at one or more selected depths.

- The distribution of the Young's modulus is plotted for the two reference depths (Fig 6.b and c)

TECHNICAL CORRECTIONS

I am not suggesting specific text amendments, but I feel that letting a native English speaker review the final manuscript would be beneficial.

- Following your suggestion, the manuscript has been sent to English language editing

[revised manuscript text omitted]

---

## Author Comment (AC2) · 27 Apr 2020

The authors are grateful for your comments and good words. We considered your suggestions and modified the paper accordingly. The answers to your comments are provided hereafter.

This paper presents a methodology for reliability analysis of complex structures and applies it to offshore wind turbines. The paper is well written for most part and presents relevant information. I have the following suggestions for improving the manuscript:

-Reference or background information for the chosen stochastic input variable parameters would be needed. -Justification for choosing the 2nd-order polynomial response function would be helpful.

- The chosen stochastic input variables are the inputs needed for the soil model (soil stiffness, cyclic contour diagrams and loads). This is made now clearer in section 2.3.

- Regarding the 2nd-order polynomial the fitting was based on a preliminary visual inspection of the outputs from the FE analysis. And the second order polynomial was also chosen because it was giving the highest R-squared. This explanation is added to the text.

-The discussion about target reliability index on pages 3and 4 needs changes. How can it be argued that SLS target should be in the range given, when Eurocode specifies it as 2.9?

- As in the Eurocodes a lower reliability level can be accepted for irreversible SLS than for ULS also for geotechnical failure modes for offshore wind turbines. (IEC 61400-1 2019) specifies the target reliability index for ULS equal to 3.3 which corresponds to probability of failure $5.10^{-4}$. The Eurocodes indicate a target annual reliability index equal to 2.9 for irreversible SLS. (IEC 61400-1 2019) does not indicate a target reliability level for SLS. Thus, conservatively we consider the target reliability index to be in the range 2.9-3.3, and conservatively use the same reliability level as for ULS, i.e. an annual reliability index equal to 3.3.

-The reliability analysis results presented in Section 3.4 need to be improved, some discussion on the design values of the random parameters would be needed. -The paper contains a number of typos.

- The design values are derived from the section 3.2. A reference to this section is now inserted in the paper.

- The manuscript has been sent to English language editing and the typos are corrected

[revised manuscript text omitted]

---

## Author Response (AR2)

The authors are grateful for your comments. We considered your suggestions and modified the paper accordingly. The answers to your comments are provided hereafter.

- although well-known in the geotechnical community, the Hardening Soil model should be accompanied by a relevant reference. Recapping - briefly - the main features/capability of the model would also be useful, for instance to explain how it differs from the simpler Mohr-Coulomb (perfectly plastic) model.

  A briefly difference regarding the modelling of the stress-strain behaviour is added. The PLAXIS reference is also inserted where a deeper explanation is presented.

- I do not agree with the new sentence "The PISA project (Byrne et al. 2019) highlights that additional soil reaction curves components (distributed moment, horizontal base force and base moment) are needed in conjunction with the p-y curves in order to have a more accurate soil structure interaction behaviour. For the present case study, the only use of the p-y curve extracted from FE is considered satisfactory". The importance of additional soil reaction components is largely driven by the aspect ratio (L/D) of the pile. As such there is no reason to believe that such components will become less relevant in the transition from monotonic to cyclic loading. In my view, neglecting "a priori" some reaction components in the analysis of short monopiles can only be justified "for the sake of simplicity".

  The text is changed according to your suggestion. For simplicity just the p-y curves were used in the aeroelastic model to derive the loads on the structure. However, it could be interesting to investigate which is the influence on the loads when employing a full PISA soil reaction curves in the aeroelastic model.

- I final check for typos should be run. See, e.g., "modulo" instead of "modulus" (twice on page 13), the full-stop after (Pisano, 2019) (line 37), the spelling of Pisano in the reference list (Pisanò is the right spelling).

  Thanks for noticing that. We went through again the text to spot other typos.

[revised manuscript text omitted]